# Photosynthesis-dependent $H_2O_2$ transfer from chloroplasts to nuclei provides a high-light signalling mechanism

Marino Exposito-Rodriguez [1,2], Pierre Philippe Laissue[1], Gabriel Yvon-Durocher [3], Nicholas Smirnoff [2] & Philip M. Mullineaux [1]

Chloroplasts communicate information by signalling to nuclei during acclimation to fluctuating light. Several potential operating signals originating from chloroplasts have been proposed, but none have been shown to move to nuclei to modulate gene expression. One proposed signal is hydrogen peroxide ($H_2O_2$) produced by chloroplasts in a light-dependent manner. Using HyPer2, a genetically encoded fluorescent $H_2O_2$ sensor, we show that in photosynthetic *Nicotiana benthamiana* epidermal cells, exposure to high light increases $H_2O_2$ production in chloroplast stroma, cytosol and nuclei. Critically, over-expression of stromal ascorbate peroxidase ($H_2O_2$ scavenger) or treatment with DCMU (photosynthesis inhibitor) attenuates nuclear $H_2O_2$ accumulation and high light-responsive gene expression. Cytosolic ascorbate peroxidase over-expression has little effect on nuclear $H_2O_2$ accumulation and high light-responsive gene expression. This is because the $H_2O_2$ derives from a sub-population of chloroplasts closely associated with nuclei. Therefore, direct $H_2O_2$ transfer from chloroplasts to nuclei, avoiding the cytosol, enables photosynthetic control over gene expression.

[1] School of Biological Sciences, University of Essex, Wivenhoe Park, Colchester CO4 3SQ, UK. [2] Biosciences, College of Life and Environmental Sciences, University of Exeter, Geoffrey Pope Building, Stocker Road, Exeter EX4 4QD, UK. [3] Environment and Sustainability Institute, University of Exeter, Penryn, Cornwall TR10 9EZ, UK. Correspondence and requests for materials should be addressed to N.S. (email: N.Smirnoff@exeter.ac.uk) or to P.M.M. (email: mullin@essex.ac.uk)

The light intensity to which plants are exposed varies seasonally and over shorter time scales due to movements of the sun, clouds and neighbouring plants. There are corresponding changes in the photosynthetic apparatus that balance energy input with photosynthetic capacity and prevent photo-oxidative damage by excess excitation energy[1]. These acclimation processes extend from rapid changes in the conformation of light harvesting complexes and dissipation of excess light as heat (non-photochemical quenching) to longer term adjustments to chloroplast composition, metabolism and antioxidant systems that require co-ordination of chloroplast and nuclear gene expression[1–4]. Exposure to high light (HL) causes rapid changes in nuclear gene expression in a photosynthesis-dependent manner[5–7] and is associated with chloroplast-to-nucleus (retrograde) signalling[1, 7, 8]. Retrograde signalling has been extensively studied in the context of light-induced chloroplast development[9]. However, the operating signals involved in response to fluctuating light intensities in mature leaves may be distinct from those involved in chloroplast development[10]. Given the complexity of the system, it is not surprising that diverse molecules that can potentially initiate and/or transduce retrograde signals have been identified. These include plastoquinone redox state, β-cyclocitral, methylerythritol cyclodiphosphate, 3′-phosphoadenosine 5′-phosphate, triose phosphates and mobile transcription factors (e.g., WHIRLY1)[5, 6, 8]. This multiplicity of signals and pathways act together to deal with a diversity of light environments in conjunction with the many other abiotic and biotic factors that must be integrated into responses that control specific sets of genes[5, 8, 11].

Chloroplasts produce various forms of reactive oxygen species (ROS) during photosynthesis and these have been proposed to activate or participate in retrograde signalling[7, 8]. One of the most reactive ROS in chloroplasts is singlet $O_2$ ($^1O_2$), which is produced by energy transfer from excited triplet state chlorophyll to $O_2$, mainly in photosystem II (PSII). Irreversible photo-inhibition is a result of oxidative damage to PSII caused by increased $^1O_2$ production[7], but this also signals the altered transcription of a set of $^1O_2$-responsive genes associated with induction of localised programmed cell death, induction of antioxidant defences and consequently, increased resistance to HL and other abiotic and biotic stresses[7, 8, 12].

The other major source of ROS is from the photo-reduction of $O_2$ at PSI (the Mehler reaction). The initial product is the superoxide anion, which dismutates to hydrogen peroxide ($H_2O_2$), catalysed by thylakoid-bound and stromal superoxide dismutases[13]. $H_2O_2$ could cause damage by oxidising a variety of macromolecular targets, including those Calvin–Benson Cycle enzymes that are regulated by the thioredoxin system[14]. It is reduced in reactions catalysed by 2-Cys peroxiredoxin and ascorbate peroxidase (APX)[15]. The regeneration of ascorbate oxidised by APX is coupled to the thiol antioxidant glutathione (GSH) and NADPH in the water-water cycle[13]. However, despite an extensive chloroplast antioxidant system, $H_2O_2$ does move out of isolated chloroplasts in vitro[16], possibly facilitated by aquaporins[17]. The mobility of $H_2O_2$ provides an opportunity for it to act as a transducer as well as an initiator of retrograde signalling. In support of this proposal, sets of genes whose expression is responsive to HL substantially overlap with those responsive to $H_2O_2$[6, 11, 18].

Photosynthesis produces $H_2O_2$ in peroxisomes because of photorespiration[19] and catalase, which is abundant in peroxisomes, has an important role in removing $H_2O_2$[20]. The involvement of $H_2O_2$ in signalling therefore raises questions about its specificity and site of action in the face of multiple origins in chloroplasts, mitochondria and peroxisomes as well extracellularly via plasma membrane localised NADPH oxidases[19]. Another problem for any retrograde signalling pathway is that plant cells harbour multiple chloroplasts, which may not respond to external stimuli in a uniform manner[5] and can move away from incident HL to increase self-shading[5, 21]. This raises the question of how a nucleus can integrate signalling from chloroplasts in different physiological states.

To address these issues and to determine if $H_2O_2$ can be a mobile retrograde signal in vivo, we used the fluorescent, genetically encoded $H_2O_2$ biosensor HyPer2[22], a newer version of HyPer[23] with expanded dynamic range. This biosensor has superior spatial, temporal and chemical resolution to the commonly used small molecule probes such as the dihydrofluorescein-related dyes[24, 25]. HyPer2 is a circular permuted (cp)YFP containing the $H_2O_2$ binding domain of E.coli OxyR, a $H_2O_2$-sensitive transcription factor[22]. The OxyR $H_2O_2$ binding domain harbours a pair of cysteine residues that are highly specific for oxidation by $H_2O_2$, resulting in formation of a disulphide bridge which in turn produces a large conformational change in the protein. When incorporated into cpYFP, the conformation change results in altered fluorescence that can be measured ratiometrically. Following oxidation, the reduction of HyPer2 is most likely dependent on glutaredoxin/GSH[22], so the probe indicates the equilibrium between $H_2O_2$-dependent oxidation and reduction by the thiol system. We use HyPer2 targeted to specific subcellular compartments to investigate the dynamics of HL-induced $H_2O_2$ production and show that chloroplast-sourced $H_2O_2$ appears in the nucleus where it can influence gene expression. To control for the pH sensitivity of HyPer2, we have used an $H_2O_2$-insensitive variant (SypHer) and an mKeima-based pH probe pHRed[26], which have also enabled us to visualise the dynamics of photosynthesis-associated pH changes in the chloroplast and cytosol.

## Results

**HyPer2 reports $H_2O_2$ in *Nicotiana benthamiana* epidermis.** We have found that the expression of HyPer2 is strongly susceptible to silencing in stably transformed *Arabidopsis thaliana* (*A. thaliana*) but is well expressed in *Nicotiana benthamiana* abaxial epidermal cells following Agro-infiltration[25]. To investigate the subcellular dynamics of $H_2O_2$ production in response to HL, we targeted HyPer2 to various subcellular compartments in *N. benthamiana*. Confocal laser scanning microscopy (CLSM) showed that the probe without specific targeting sequences (cHyPer2) was located in the cytosol and nuclei (Supplementary Fig. 1a). Probes with chloroplast stroma (sHyPer2) and nuclear (nHyPer2) targeting sequences were found in their appropriate subcellular locations (Supplementary Fig. 1b, c). HyPer2 is a ratiometric probe showing an increase in fluorescence emission at 520 nm when excited at 488 nm compared to 405 nm (F488/405). To confirm that HyPer2 responds to $H_2O_2$ in epidermal cells, F488/405 was measured by CLSM. F488/F405 increased when the probe was oxidised by $H_2O_2$ addition in all three compartments but most strongly in the cytosol (Fig. 1). Imposition of reducing conditions by dithiothreitol (DTT) decreased the ratio to the same low value in all the compartments (Fig. 1a). The F488/405 ratio in the cytosol increased 7-fold from reduced to oxidised (Fig. 1b), which is the same value as reported for HyPer2 expressed in HeLa cells[23]. Therefore 10 mM external $H_2O_2$ is sufficient to fully oxidise HyPer2 in the cytosol but not the chloroplast stroma. We conclude that HyPer2 is an effective $H_2O_2$ probe in *N. benthamiana* cells. The greater oxidation state in the cytosol compared to the stroma could reflect either limited $H_2O_2$ penetration or a greater antioxidant capacity in the chloroplasts.

To use *N. benthamiana* epidermal chloroplasts for investigating responses to high light, it was necessary to confirm that they are photosynthetic. Imaging of chlorophyll fluorescence quenching in epidermal cells in response to a range of light intensities indicated a similar PSII maximum efficiency to that of mesophyll cells (Supplementary Fig. 2a, b). HL (1000 μmol photons m$^{-2}$ s$^{-1}$) treatment of leaves was used to investigate $H_2O_2$ dynamics. The plants for these experiments were grown at 120 μmol photon m$^{-2}$ s$^{-1}$ (growth light; GL) and in response to 1 h of HL the maximum dark-adapted PSII quantum efficiency ($F_v/F_m$) decreased from 0.7 to 0.5 (Supplementary Fig. 2c, d). This was associated with an approximately 50% increase in foliar $H_2O_2$ content (Supplementary Fig. 2e). However after 24 h in GL, $F_v/F_m$ almost fully recovered to pre-HL values (Supplementary Fig. 2c, d). Therefore, the HL treatment used in this study caused only a transient photoinhibition of photosynthesis and would have produced insignificant oxidative damage to PSII.

Photosynthetically active chloroplasts show a rapid rise in the pH of the stroma upon illumination[27]. Therefore, we targeted the pH sensor pHRed[26] to the chloroplast stroma of epidermal cells. pHRed was chosen because its pKa (6.6) and >10-fold dynamic range[26] would be most effective at reporting the anticipated alkalinisation of the stroma. During the dark/light transition cycles, the changes in fluorescence emission of stromal pHRed (spHRed) from its two excitation maxima (F543/458) indicated a rapid alkalinisation (~2 min) of the stroma, as predicted by light-dependent proton transport into the thylakoid lumen, returning to the initial values during the dark phase[27] (Supplementary Fig. 3a, b). In addition, the rapid alkalinisation was inhibited by the photosynthetic electron transport inhibitor 3-(3,4-dichlorophenyl)-1,1-dimethylurea[28] (DCMU; Supplementary Fig. 3c, d). Our measurements are all indicative of active photosynthesis within the epidermal chloroplasts.

**Light-induced pH changes in the chloroplast stroma**. The HyPer2 constructs were used to follow the subcellular dynamics of $H_2O_2$ production in response to HL. Initial measurements with

sHyPer2 showed a steady increase in F488/F405 over 20 min, which then decreased in a biphasic manner in the dark (Supplementary Fig. 4a). However, interpretation of data from HyPer or HyPer2 is complicated by the pH sensitivity of the probe[22, 23]. Since the F488/F405 ratio for sHyPer2 could increase with pH as well as oxidation, we compared in parallel the time course of pH changes using spHRed (Supplementary Fig. 4b) using a bicistronic plasmid vector harbouring chimeric genes coding for both probes. Within 2 min of HL exposure, stromal alkalinisation caused a decrease in the F543/458 ratio of spHRed. This could mean that in the first 2 min of HL, the change in sHyPer2 fluorescence (Supplementary Fig. 4b) is driven by the rapid rise in pH. Therefore, we concluded that the steady increase in HyPer2 F488/405 ratio thereafter was caused by oxidation of the probe. Furthermore, sHyPer2 was also effectively re-reduced in the dark (Supplementary Fig. 4c), presumably by the chloroplast thiol system.

For all subsequent experiments, we constructed an $H_2O_2$-insensitive but still pH-sensitive HyPer2 mutant (SypHer2) by converting cysteine 199 to serine in the OxyR active site[29]. In all subsequent time course experiments we used SypHer2, targeted to the appropriate subcellular location, to estimate any impact of pH changes on HyPer2 fluorescence. SypHer2 is ideal as a pH control for HyPer2 since they have identical pH responses (pKa = 8.6).

**The dynamics of $H_2O_2$ production in response to HL**. After HL illumination, the F488/F405 ratio of sHyPer2 increased in repeated experiments over 60 min (Figs. 2a and b). This increase can be attributed to oxidation by $H_2O_2$ since the F488/F405 ratio of stromal SypHer (sSypHer2) did not increase over the same period (Figs. 2a and b). To determine the differences between HL response time courses we fitted response curves of F488/F405 following HL exposure using a generalised additive mixed effects model (GAMM). This procedure demonstrated that the best fitting model included different forms for the time-series for sHyPer2 and sSypHer2 (Supplementary Fig. 5a, Supplementary

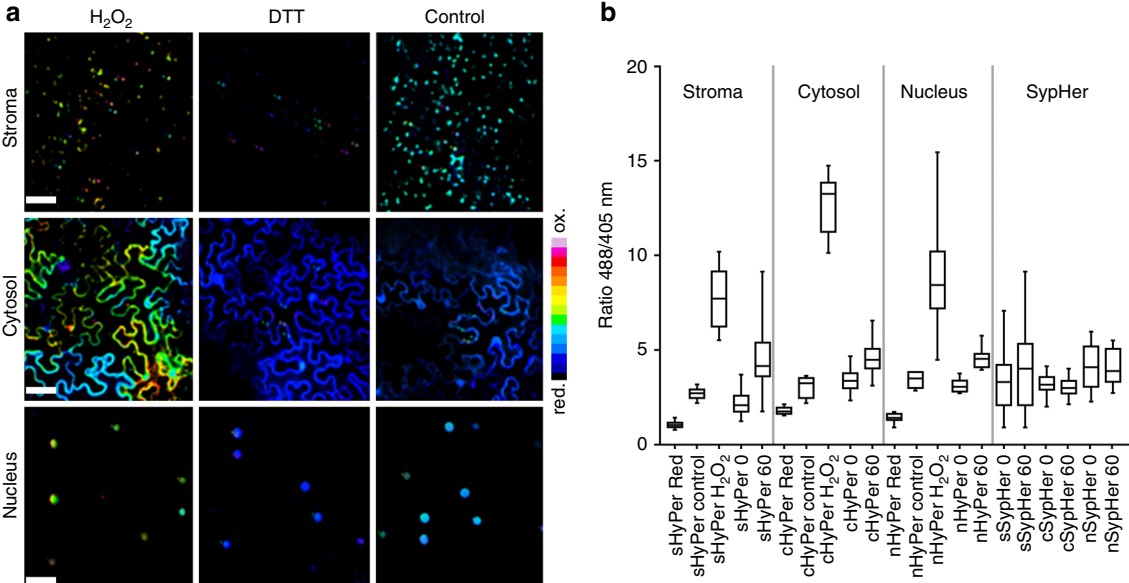

**Fig. 1** HyPer2 responds to hydrogen peroxide and dithiothreitol as well as high light in different subcellular compartments of *Nicotiana benthamiana* epidermal cells. **a** Ratiometric images (F488/405 nm) of fluorescence emission at 530 nm following excitation at 488 and 405 nm showing oxidised state (10 mM $H_2O_2$ for 20 mins), reduced state (10 mM DTT for 20 mins) and resting state (Control) of stromal, cytosolic and nuclear targeted HyPer2 respectively. Oxidation of HyPer2 increases the 488/405 nm ratio. **b** Comparison of the response of HyPer2 and SypHer to reduction by DTT, oxidation by $H_2O_2$ and exposure to 60 min HL (1000 μmol m$^{-2}$ s$^{-1}$) and at the start (0 min; GL conditions). Scale bar = 50 μm. Error bars represent the SD, $n$ = 11 cells (cHyPer2); 23 chloroplasts (sHyper2) and 11 nuclei (nHyPer2) per treatment from 3 independent experiments

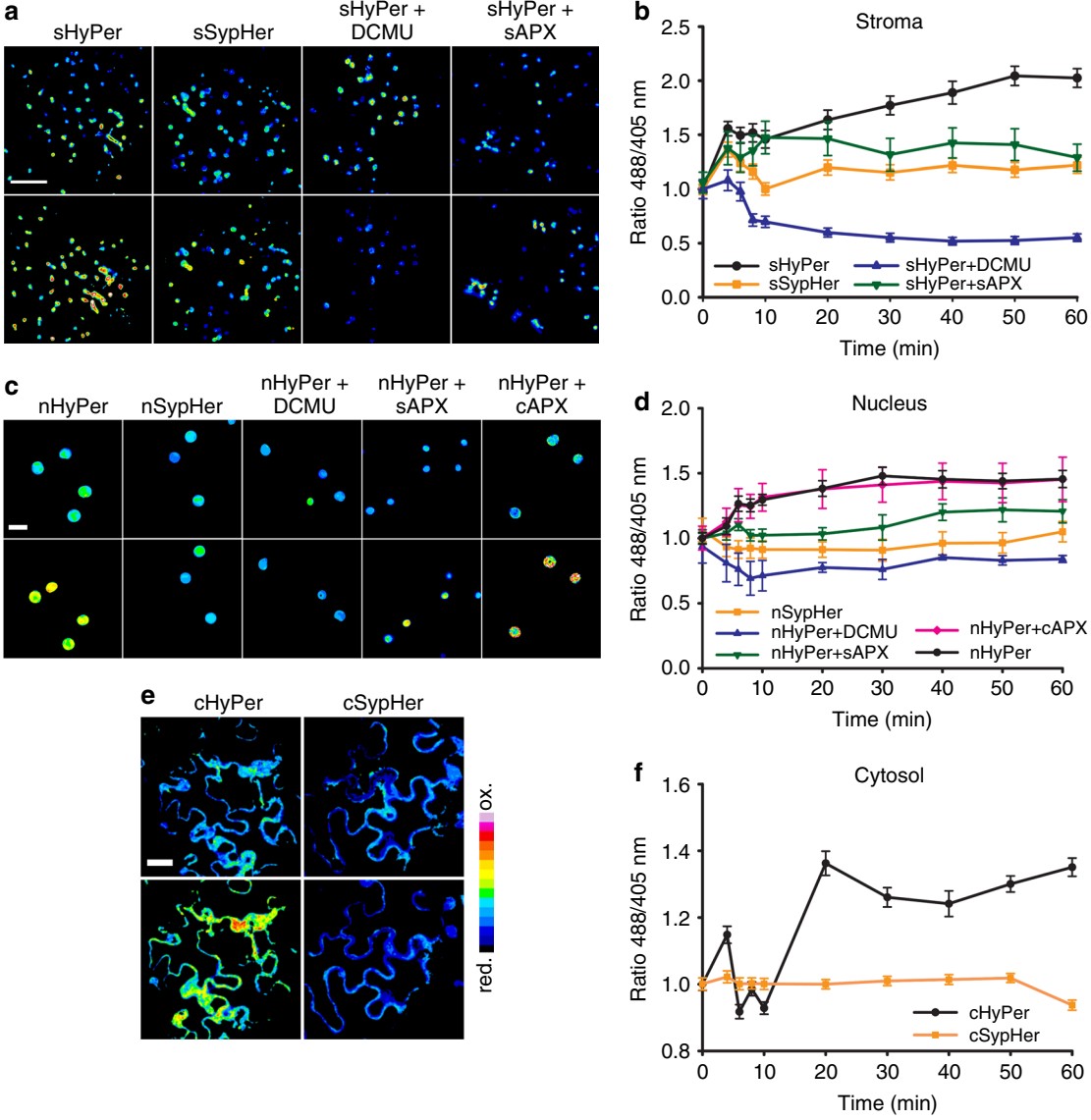

**Fig. 2** Time course of HyPer2 and SypHer2 responses to HL expressed in different subcellular compartments of *N. benthamiana* epidermal cells.
**a**, **b** *N. benthamiana* abaxial epidermal cells expressing sHyPer2 and sSypHer2. **a** 488/405 nm ratio images of resting state in the dark (*upper row*) and after 60 min HL (*lower row*). From left to right, the panels show the fluorescence of chloroplasts of cells from leaves expressing sHyPer2, sSypHer2, sHyPer2 treated with DCMU (10 μM for 20 min prior to illumination) and over-expressing sAPX, respectively **b** The effect of exposure to HL over 60 min on the 488/405 nm fluorescence ratio of chloroplasts in epidermal cells expressing sHyper2 and sSypHer 2. The data points were normalised to the mean initial value for each treatment and represent the mean ± SEM values from 40–70 chloroplasts from 3 independent experiments per treatment. **c**, **d** Equivalent to panels a and b except that the cells expressed nHyPer2 and nSypHer2. From left to right, the panels show the fluorescence from nuclei expressing nHyPer2, nSypHer2, sHyPer2 treated with DCMU (10 μM for 20 min prior to illumination), over-expressing sAPX and cAPX, respectively. The data points were normalised to the mean initial value for each treatment and represent the mean ± SEM from 8–18 nuclei from 3 independent experiments per treatment. **e**, **f** *N. benthamiana* abaxial epidermal cells expressing cHyPer2 and cSypHer2. **e** 488/405 nm ratio images of resting state in the dark (upper row) and after 60 min HL (lower row) of cHyPer2 and cSypHer2 right and left column, respectively. **f** Quantification of the HL-dependent changes in cHyPer2 and cSypHer 2 fluorescence over 60 min HL. The data points were normalised to the mean initial value for each treatment and represent the mean ± SEM from 12–15 cells from 3 independent experiments per treatment. Scale bar = 50 μm

Table 1). The detection of light-dependent $H_2O_2$ in the chloroplasts is consistent with previous observations using chemical probes[11, 16].

By comparing the response curves of HyPer2 and SypHer2 in the nuclei (Figs. 2c, d) and cytosol (Figs. 2e, f) we concluded that HL also causes an increase in $H_2O_2$ over a similar time course in these compartments. Indeed, the best fitting GAMMs to the changed in F488/F405 in both the cytosol and nucleus included different smooth functions for the time-series of HyPer2 and SypHer2 (Supplementary Fig. 5b, c; Supplementary Table 1).

To determine the origin of $H_2O_2$ in the nuclei, we used DCMU to inhibit photosynthetic electron transport. This treatment not only decreased HyPer2 oxidation in the stroma (Figs. 2a, b) but also significantly decreased HyPer2 oxidation in nuclei (Figs. 2c, d), as evidenced by reduced median F488/F405 values in the DCMU treatments in the best fitting models for F488/F405 dynamics in both the stroma and nucleus (Supplementary Fig. 5a, b; Supplementary Table 1). These results show that the HL-mediated increase in nuclear $H_2O_2$ is dependent on photosynthetic electron transport.

**Stromal ascorbate peroxidase influences $H_2O_2$ in the nucleus.** We reasoned that the HL-dependent increase in nuclear $H_2O_2$ could derive from oxygen photo-reduction in the chloroplast[7].

However, substantial amounts of $H_2O_2$ are produced in peroxisomes from photorespiration via glycolate oxidase in HL-exposed leaves[19]. Therefore, the rapid increase in oxidation of nHyPer2

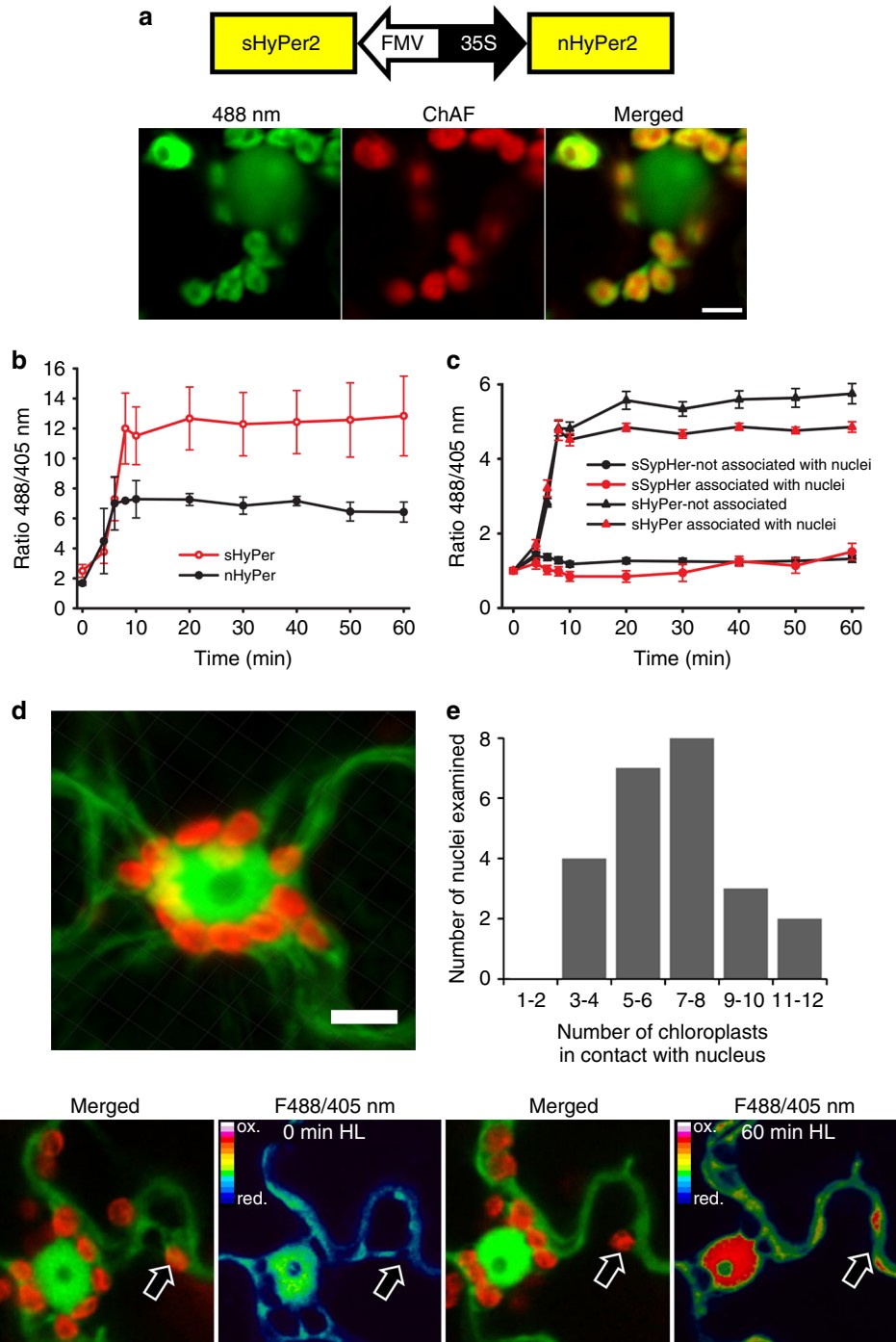

**Fig. 3** High light causes coincident oxidation of HyPer2 in chloroplasts and nuclei. **a** Expression of sHyPer2 and nHyPer2 from a bicistronic construct. The left, middle and right panels show HyPer2 fluorescence from 480 nm excitation light, chlorophyll auto-fluorescence (ChAF) and a combined image of the two respectively. Scale bar = 5 μm. **b** The effect of exposure to HL over 60 min on the 488/405 nm fluorescence ratio of chloroplasts in epidermal cells expressing sHyPer2 and nHyPer2. The 488/405 nm ratio is normalised to 1 at 0 min for each dataset. The data points represent the mean ± SEM values from 40 chloroplasts and 3 nuclei from 3 independent experiments. **c** From the same experiment as in **b** but sHyPer2 488/405 nm ratios are plotted for individual chloroplasts classified as appressed or detached from nuclei. **d** Typical higher magnification of a cell expressing cHyPer2 fluorescence from 480 nm excitation light showing the position of chloroplasts revealed by chlorophyll auto-fluorescence. Scale bar = 5 μm. **e** The distribution of chloroplasts numbers associated with nuclei. 25 nuclei were examined. **f** Higher magnification of ChAF and 480/405 nm ratio of a cell expressing cHyPer2. Panel shows the merged images of cHyPer2 (F480 nm) with ChAF and the ratio 480/405 nm of cHyPer2, in the dark (first and second images) and after 60 min HL (third and fourth images), respectively. The discrete zones of cytosol-located oxidised HyPer2 (arrowed in right hand panels) detected in the proximity of chloroplasts that are not associated with nuclei. Scale bar = 10 μm

could be a consequence of $H_2O_2$ sourced from peroxisomes. To address this issue, we co-expressed an Arabidopsis stromal ascorbate peroxidase (sAPX; At4g08390) with sHyPer2 or nucleus-targeted HyPer2 (nHyPer2). The sAPX was over-expressed to increase the capacity of the chloroplasts to remove $H_2O_2$ using ascorbate as reductant. To ensure that sAPX and sHyPer2 or nHyPer2 were expressed in the same cells, we used a bi-cistronic Ti vector. The correct subcellular localisation of sHyPer2 and nHyPer2 was confirmed by CLSM (Supplementary Fig. 6a–c) and a consequent 5-fold increase in APX enzyme activity was measured in leaf extracts (Supplementary Fig. 6d). Expression of this construct attenuated HyPer2 oxidation in both chloroplast stroma and nuclei. This was evidenced by reduced median F488/F405 values in the sAPX relative to the sHyPer2 and nHyPer2 treatments in the best fitting models for F488/F405 dynamics in both the stroma and nucleus (Figs 2a, b; Supplementary Fig. 5a, b; Supplementary Table 1). This result demonstrates that the HL-dependent nuclear $H_2O_2$ is largely derived from the chloroplasts. Overall, these results strongly suggest a transfer of photosynthetically produced $H_2O_2$ directly from the chloroplasts to nuclei although a contribution from the peroxisomes cannot be discounted.

Over-expression of an Arabidopsis cytosolic APX (APX1; At1g07890; Supplementary Fig. 6c, d) had little effect on the oxidation state of nHyPer2, with similar time-series and median values of F480/F405 ratios in nHyPer2 and nHyPer2 + cAPX treatments (Fig. 2b; Supplementary Fig. 5b; Supplementary Table 1). This was in contrast to sAPX expression in the stroma, which exhibited marked different time-series and median values of F480/F405 ratios in nHyPer2 and sAPX treatments under HL (Figs. 2c, d). This result suggests that the transfer of $H_2O_2$ to the nucleus from chloroplasts does not involve extensive transit though the cytosol. To compare the time course of stromal and nucleus $H_2O_2$ increase in the same cells, a bicistronic Ti vector was made, which allowed simultaneous monitoring of HyPer2 fluorescence in both subcellular compartments (Fig. 3a). The response times of the two compartments could not be resolved at our 2 min time resolution (Fig. 3b). This confirmed that the movement of $H_2O_2$ from chloroplasts to the nucleus is rapid and likely involves diffusion between the two compartments rather than a signalling pathway that generates new nuclear $H_2O_2$.

N. benthamiana abaxial epidermal cells contain relatively few chloroplasts compared with mesophyll cells[30]. In this experiment expressing both sHyPer2 and nHyPer2 both nuclei and chloroplasts displayed HyPer2 fluorescence, and we observed that there were chloroplasts spread throughout the cytosol as well as some associated with nuclei (Fig. 3a and Supplementary Fig. 1a). A median value of 7–8 chloroplasts were associated with each nucleus (Figs. 3d, e). The rates of oxidation of sHyPer2 under HL in these two sub-populations of chloroplasts were similar over the first 10 min of HL (Fig. 3c). However, the steady-state oxidation of sHyPer2 was lower in chloroplasts associated with nuclei compared with those that were not (Fig. 3c). The HL steady state HL response of sSypHer did not differ between

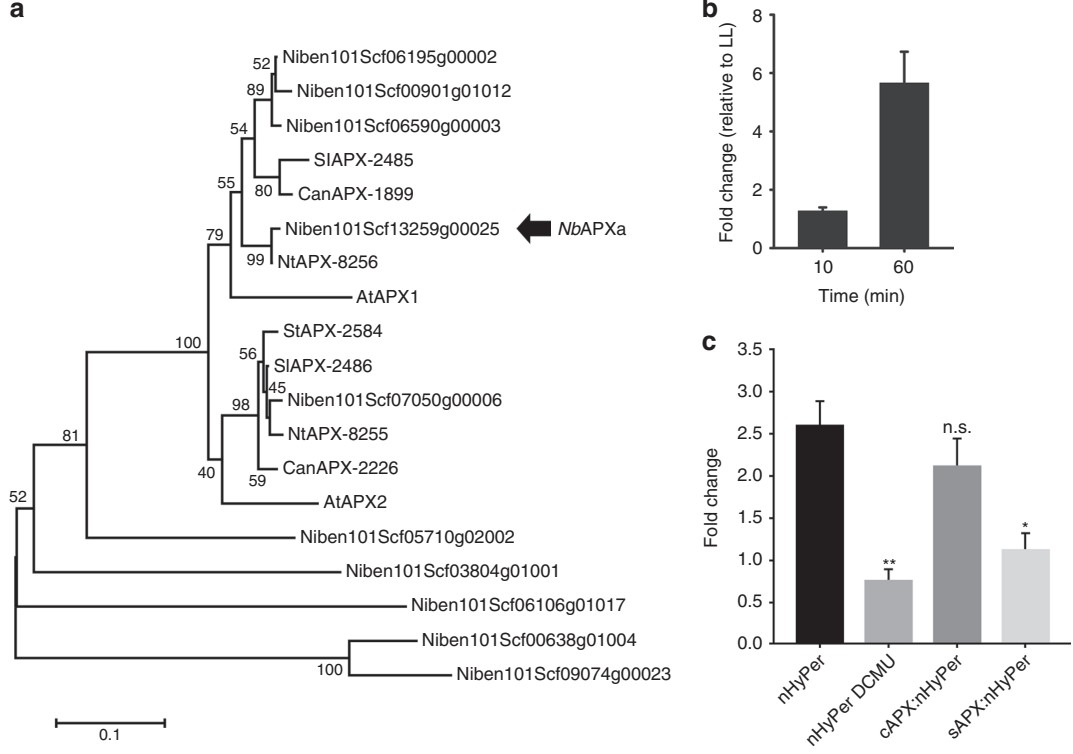

**Fig. 4** The HL-responsive expression of an *APX* gene is dependent upon $H_2O_2$ from the chloroplasts. **a** Neighbour-joining tree based on predicted amino acid sequences showing the phylogenetic relationships of putative *APX* genes from *N. benthamiana* (NiBen) Solanum lycopersicum (Sl), (Can, *Capsicum annuum*), Nt (*Nicotiana tabacum*), St (*Solanum tuberosum*). Sequences were extracted from PeroxiBASE. Numbers next to the names indicate the identifier in the database. The *APX* gene shown to be HL responsive is marked (*Nb*APXa). Bootstrap values (1000 replicates) are displayed near the nodes. The scale indicates the number of nucleotide substitutions per site. **b** Steady state levels of *NbAPXa* RNA leaves of *N. benthamiana* after 10 min and 60 min exposure to HL relative to a low light (LL) control. **c** Steady state levels of *NbAPXa* RNA leaves of *N. benthamiana* after 60 min exposure to HL relative to a LL controls. All leaves expressed nHyPer2 and a proportion of them were additionally treated with DCMU ($10\,\mu M$ for 20 min prior to illumination) or co-expressed sAPX or cAPX. The nHyPer2 expressing LL controls were treated in the same way. The values are the means ± SEM ($n = 8$) plants from two experiments. Significant differences (one way ANOVA) between the HL exposed leaves and those also expressing sAPX or treated with DCMU are denoted by **($P = 0.004$) and *($P = 0.01$); n.s. is not significant ($P > 0.05$)

chloroplast populations so the difference in sHyPer oxidation was not caused by pH differences. Oxidation of cytosol-located HyPer2 was detected in the proximity of both chloroplast populations during HL exposure (Fig. 3f). The persistence of chloroplast-sourced $H_2O_2$ in the cytosol is severely restricted by the antioxidant capacity of the cytosol, which is rapidly enhanced in response to HL[31, 32]. Direct transfer of $H_2O_2$ from the chloroplasts into the nucleus means that it avoids scavenging by the cytosol antioxidant system. There is also a possibility that the nuclear $H_2O_2$ is produced in situ as a result of stimulation by chloroplast-sourced $H_2O_2$ which is discussed later.

**Chloroplast-sourced nuclear $H_2O_2$ affects gene expression.** HL and $H_2O_2$ treatments are well known to increase the expression of specific and overlapping sets of genes in Arabidopsis (see Discussion). The photosynthesis-dependent nuclear $H_2O_2$ accumulation may provide a signal to mediate induction of HL-responsive genes. HL-induced gene expression in *N. benthamiana* has not been studied, so we searched for candidate HL-inducible nuclear genes with which to test for a signalling role for nuclear chloroplast-sourced $H_2O_2$. Our starting point was to test the expression of *N. benthamiana* (Nb) *cAPX* genes in response to HL. This was because the Arabidopsis *cAPX* homologues have long served as marker genes for studying HL signalling in this species[11, 31, 33]. BLAST alignments with Arabidopsis APX1 and APX2 derived protein sequences identified a candidate *NbAPX* gene, which we named *NbAPXa* (Fig. 4a). Exposure of leaves of *N. benthamiana* to 1 h HL treatment resulted in a 6-fold increased expression of *NbAPXa* (Fig. 4b). Since DCMU and sAPX over-expression both attenuated the nuclear $H_2O_2$ increase in HL (Figs. 3c, d), we investigated their effect on the expression of *NbAPXa* (Fig. 4c). Both treatments substantially decreased HL-inducible expression of this gene. In contrast, over-expression of a *cAPX* did not significantly decrease *NbAPXa* expression (Fig. 4c). These results support the conclusion that the increase in chloroplast-sourced, directly transferred nuclear $H_2O_2$ acts as a signal to influence nuclear gene expression in response to HL.

## Discussion

Using the $H_2O_2$ specific probe HyPer2 targeted to chloroplasts and nuclei we show that chloroplast-sourced $H_2O_2$ is transferred to nuclei where it may act as a signal to induce HL-responsive gene expression. HyPer2 successfully detected an increase in $H_2O_2$ in the chloroplast stroma after illumination with HL as seen by a steady increase in probe oxidation over ~20 mins. On transfer from dark to light, stromal pH increases to around pH 8[27]. Since HyPer2 F488/405 ratio increases with probe oxidation and higher pH[34] it is important to separate these effects. Therefore, we measured pH changes in the stroma using a $H_2O_2$-insensitive HyPer2 variant (SypHer2) and pHRed, a recently developed pH sensor[26]. Both these sensors showed that pH adjusted within 3 mins to the illumination and therefore the subsequent sustained ratio increase with HyPer2 can be assigned to probe oxidation. DCMU prevented HyPer2 oxidation. Since DCMU blocks the $Q_B$ plastoquinone (PQ) binding site of PSII, electron transfer is blocked, PQ becomes oxidised and electron transport to PSI decreases[28]. The results show that the primary source of $H_2O_2$ in the chloroplasts is therefore due to electron transport beyond PSII and, in agreement with previous studies, is most likely mediated by the Mehler reaction[7, 16]. An alternative explanation for the origin of nuclear $H_2O_2$ is that chloroplast-sourced $H_2O_2$ production activates an $H_2O_2$ generating system in nuclei. Our results cannot completely exclude this possibility but secondary $H_2O_2$ production would most likely incur a time lag

and, at least within the 2 min time resolution of the imaging method, the time courses of stromal and nuclear HyPer2 oxidation were indistinguishable (Fig 3b). $H_2O_2$ diffusion over this spatial scale would take less than one second. Even if nuclear $H_2O_2$ is secondary, our essential conclusion that chloroplast-sourced $H_2O_2$ increases nuclear $H_2O_2$ is not invalidated. In contrast to its effect on $H_2O_2$ production, DCMU plus HL increased the oxidation of roGFP2 targeted to Arabidopsis chloroplasts[35]. Therefore, the increased $^1O_2$ production in PSII caused by DCMU[12], rather than $H_2O_2$, oxidises the chloroplast thiol system, since roGFP2 is responsive to the oxidation state of glutathione[36].

Our key observation is that HL caused oxidation of HyPer2 in the cytosol and also the nuclei in a photosynthetic electron transport-dependent manner. Therefore, photosynthesis-sourced $H_2O_2$ could influence nuclear redox state and gene transcription. The expression of sAPX in chloroplasts very effectively decreased HyPer2 oxidation in the nuclei. This shows that there is an important role for the ascorbate-APX system in controlling the transport of $H_2O_2$ out of chloroplasts, probably via aquaporins[17]. The inhibition of nuclear $H_2O_2$ production by DCMU shows that it is derived from photosynthesis, while the effectiveness of sAPX in decreasing nuclear $H_2O_2$ shows that it is largely derived from the chloroplasts, not from photorespiratory $H_2O_2$ produced by glycolate oxidase in the peroxisomes. Expression of cAPX had virtually no effect on nuclear HyPer2 oxidation and possibly this is because $H_2O_2$ moves directly from chloroplasts that are tightly associated with nuclei thus avoiding reduction by an enhanced APX activity in the cytosol. Expression of an APX (*NbAPXa*) was rapidly induced by HL and this response was inhibited by DCMU and sAPX but not cAPX (Fig. 4). Since the relative effect of these treatments on nHyPer2 was the same, there is strong evidence that the pool of $H_2O_2$ in the nucleus, directly transferred from the chloroplasts, can participate in signalling and therefore provide a spatially dependent direct retrograde signalling pathway. The lack of an effect of cAPX expression on nuclear $H_2O_2$ and NbAPXa suggests that peroxisome derived $H_2O_2$ plays a relatively minor role. Further work is needed to assess the generality of this conclusion.

Based on studies in yeast and mammalian cells, $H_2O_2$ primarily affects gene expression by oxidising redox-active transcription factors (TFs), which then alters their activity, subcellular location or marks them for degradation[37–40]. Yeast has redox-sensitive thiol-disulphide TFs (e.g., Yap1/Pap1) whose activation by oxidation is mediated by glutathione peroxidases (GPXs; e.g., Gpx3/Orp1) or peroxidases (e.g., Tsa2) that are in turn transiently oxidised by $H_2O_2$[37, 38]. It has been argued that this redox relay-mediated regulation of TFs is likely to be ubiquitous in eukaryotes and provides specificity for signalling involving $H_2O_2$ despite the considerable cellular scavenging capacity for this ROS[39, 40]. In plants, redox-regulation of TFs has not been extensively characterised. However, two heat shock transcription factors in Arabidopsis leaves subjected to stress may be regulated by the reduction/oxidation of specific cysteine residues[33, 40]. There is also evidence that a nuclear-localised TF (RAP2.4a) is redox active. It is an APETALA2 family TF that controls expression of 2-Cys peroxiredoxin and other antioxidant enzymes and mutants are sensitive to fluctuating light. RAP2.4a-GFP fusions locate to the nucleus and it has essential cysteine residues that are oxidised in vitro by $H_2O_2$[41]. RAP2.4a could therefore be a target for control of HL responses via nuclear $H_2O_2$.

$H_2O_2$ movement from chloroplasts to nuclei would be facilitated if they were closely or physically associated. cAPX over-expression had a reduced effect on HL-induced nuclear $H_2O_2$ levels compared with sAPX over-expression (Figs. 2c, d). This result would be explained if movement of $H_2O_2$ from the

chloroplasts to the nucleus involved only a minimal or no transit through the cytosol. Our observations of *N. benthamiana* epidermal cells repeatedly indicated close association of nuclei with a subset of chloroplasts. Physical association of chloroplasts with the nucleus has been reported in a wide range of species[21, 42]. The nature of the tethering between chloroplasts and nuclei is not known although the ER has been noted to form a layer between them[42]. It has been suggested that stromules (highly mobile chloroplast extensions), formed during the development of the immune hypersensitive response (HR) in *Arabidopsis* and *N. benthamiana* could transfer $H_2O_2$ directly from chloroplasts to nuclei[30]. We did not observe stromule formation during the HL response in *N. benthamiana* epidermal cells but this may be because our experiments were completed by 1 h of HL while stromule formation during the HR takes up to 46 h[30]. Peroxisomes could also provide some of the nuclear $H_2O_2$ but our results argue for a predominant role for chloroplasts because of the large effect of stromal APX in attenuating nuclear HyPer2 oxidation and NbAPXa expression compared to cytosolic APX. There is no evidence for attachment of peroxisomes to nuclei, so $H_2O_2$ from this organelle would need to transit the cytosol.

The capacity of the Mehler reaction to act directly in a photo-protective role as an electron sink/safety valve to dissipate excitation energy has been questioned due to its low capacity[43]. We suggest that the Mehler reaction, while being an inevitable side reaction of photosynthetic electron transport, has been co-opted as a signalling mechanism. Interestingly, in their assessment of the inability of the Mehler reaction to act as a safety valve in French beans, Driever et al.[43] commented that $H_2O_2$ generated by the Mehler reaction is unlikely to leave the chloroplast and act as a mobile signal because of the high antioxidant capacity in that organelle. Our results contradict this view and support the simulations of Marinho et al.[39] that show sufficient oxidation of a target $H_2O_2$ sensor can still occur when the competing $H_2O_2$ scavenging system is $10^9$ times faster than rate of oxidation of the sensor. Critically, Mubarakshina et al.[16] showed that the proportion of oxygen photo-reduction compared to total electron transport capacity increased with increasing light intensity. This indicates that $H_2O_2$ production is an effective measure of "excess excitation energy"[44] and could therefore be a useful signal to activate acclimatory responses. Investigation of a wider diversity of plants has shown a high capacity for the Mehler reaction in mosses[45], diverse microalgae[46, 47] and conifers[48]. Conifers and mosses have low photosynthetic $CO_2$ assimilation capacity and are therefore likely to be frequently exposed to excess excitation energy. Furthermore, the chloroplast origin of nuclear $H_2O_2$ would allow this mechanism to work in $C_4$ plants, which have a low rate of photorespiratory $H_2O_2$ production[43].

If chloroplast-sourced $H_2O_2$ is a signal for acclimation, then engineering plants for increased antioxidant capacity might, under some circumstances, paradoxically increase susceptibility to HL and other stresses. For example, tobacco plants with higher reduced glutathione as a result of expressing bacterial γ-glutamylcysteine synthetase in chloroplasts are more sensitive to HL, showing bleaching, structural changes to chloroplasts and decreased expression of HL-induced FeSOD[49]. Addition of exogenous GSH impairs HL-induced *APX1* and *APX2* expression and causes greater photoinhibition[31]. Furthermore, an Arabidopsis thylakoid APX mutant is more resistant to heat stress while a cAPX mutant is more susceptible[50] and double catalase/APX mutants are more resistant to oxidative stress[51]. sAPX is susceptible to inactivation by $H_2O_2$ and can be made less sensitive by removal of a 16-amino acid loop that is not present in the $H_2O_2$ resistant cytosolic and glyoxysomal enzymes[52]. Possibly, as previously speculated[53], the specific susceptibility of

sAPX could be important in facilitating $H_2O_2$ loss from the chloroplast for retrograde signalling.

In conclusion, we have used the chemical specificity and spatial resolution of HyPer2 to show that chloroplast-derived $H_2O_2$ can be detected in the nucleus. Furthermore, evidence that this could provide a direct signal in the nucleus to control gene expression via redox active TFs is discussed. This mechanism provides a very direct and highly specific means for nuclear gene expression to respond to changes in photosynthetic activity.

## Methods

**Growth conditions.** *Nicotiana benthamiana* plants were grown in controlled environment chambers under 8 h photoperiod, 22 ± 1 °C, 1 KPa vapour pressure deficit and a photosynthetically active photon flux density (PPFD) of 120 ± 10 $\mu mol\ m^{-2}\ s^{-1}$ and hereafter termed GL conditions.

**Construction of plasmids.** Gateway vectors and cloning methods were used (Invitrogen, Paisley, UK). The PCR primers designed for cloning into pENTR/D-TOPO included addition of 4-nucleotide CACC at the 5′ end of the forward primer. The HyPer-AS entry clone (Evrogen, Moscow; cat. FP943) was used to create the HyPer-fusions targeted to chloroplasts and nuclei. For chloroplast stroma targeting, the HyPer2 sequence was amplified using the primers P1 and P2 (Supplementary Table 2) to create a version without a start codon and to introduce a Hind III site at the 5′ end (coding for the N-terminus) of Hyper. For chloroplast localisation, the 240 bp plastid targeting signal from pea ribulose bisphosphate carboxylase small subunit (RbcS; ref. 54) was synthesised (Epoch Biolabs, Inc. Missouri, USA) and cloned into pBluescript SK- flanking by SpeI and HindIII at the 5′ and 3′ termini, respectively. After digestion with SpeI and HindIII, the RbcS sequence was ligated in front of the HindIII digested-HyPer. The ligated products were purified and amplified by PCR with primers P3 and P2 (Supplementary Table 2). For nuclear localisation, the HyPer2 sequence was amplified using the primer P4 and P5 (Supplementary Table 2) to create a version of HyPer2 with two nuclear localisation signals from the coding sequence of Simian virus 40 (SV40) large T antigen at its 5′ and 3′ termini. The PCR products of the chloroplast- and nuclear-targeted versions of HyPer2 were cloned into pENTR/D-TOPO following the manufacturer's procedures (Invitrogen, Paisley, UK).

HyPer2 and SypHer, the $H_2O_2$-insensitive but pH-responsive version of HyPer[29] were generated by site-directed mutagenesis (SDM) using the QuikChange Site-Directed Mutagenesis Kits (Genomics Agilent), using as template the pENTR: HyPer2 clones described above. SDM was performed using 18 reaction cycles with 5 min for elongation at each cycle and the primers were used at 10-fold the suggested concentration. For HyPer2, the A233V mutation was introduced using the primers P6 and P7 (Supplementary Table 2). Once the mutation and the absence of other mutations were confirmed by sequencing, HyPer2 and the primers P8 and P9 (Supplementary Table 2) were used to generate the SypHer2 versions of the constructs.

The pH sensor pHRed (25) in vector pRsetB (AddGene:31472) was used as template for PCR using the primers pHRed P10 and P11 (Supplementary Table 2), which generated a PCR product without specific targeting sequences. For the chloroplast targeted version, the pHRed sequence was amplified using the primer P12 and P11 (Supplementary Table 2) to create a version without a start codon and to introduce a HindIII site at the 5′ end of the pHRed coding sequence. The transit sequence of RbcS was ligated in front of the pHRed following the same strategy used with HyPer. The ligated products were purified and amplified by PCR with primers P2 and P11 (Supplementary Table 2). The sequences of APX were amplified from Arabidopsis cDNA using the primers P13 and P14 (Supplementary Table 2) for APX1 (At1g07890) and P15 and P16 (Supplementary Table 2) for sAPX (At4g08390). The PCR products described above were purified and cloned into pENTR/D-TOPO (Invitrogen, Paisley, UK).

After sequence confirmation, the resulting clones were sub-cloned using LR reaction into a Gateway T-DNA destination vector pUB-DEST[55] following the manufacturer's procedures (Invitrogen, Paisley, UK). To facilitate the co-expression of HyPer2 together with cAPX or sAPX from Arabidopsis, we used pGemini (a gift from Dr. Andrew Simkin) a bi-directional vector built in house where the expression of two transgenes is driven by back-to-back CaMV35S and Figwort Mosaic Virus (FMV) constitutive promoters. The incorporation of two independent LR-clonase sites in an anti-parallel orientation permitted the integration of two different cDNAs via a single LR-clonase event. The colonies were screened by two rounds of PCR for dual-integration of each cDNA, using the primers P17, P18 and P19 (Supplementary Table 2) targeted to CaMV35S, FMV and HyPer, respectively. The positive colonies were screened using the primers P17 and P18 in combination with P14 or P16 (Supplementary Table 2) for APX1 or sAPX respectively.

**Transient expression.** Transient expression in *N. benthamiana* leaves was used as a rapid approach to monitor the biosensors' response in different subcellular compartments[56]. The above-mentioned constructs were transformed into

Agrobacterium strain GV3101::pMP90[57]. Agrobacterium expressing the P19 protein of tomato bushy stunt virus[58] was used to suppress gene silencing. Bacterial cultures were grown overnight at 28 °C in Luria-Bertani medium, supplemented with the rifampicin and gentamycin for bacterial selection and spectinomycin, kanamycin or kanamycin plus hygromycin for plasmids derived from pUB-DEST, pBin19::p19 or pGemini, respectively. After centrifugation and washing twice in the infiltration buffer (10 mM MgCl₂, 10 mM MES, pH 5.8, 150 mM acetosyringone), the bacterial pellets were re-suspended in the same buffer and the cultures were incubated at room temperature for 2–3 h. Inoculation of the biosensor and p19 construct containing bacteria was at a density (OD$_{600 nm}$) of 0.2 and 0.04, respectively, which were achieved by dilution with infiltration buffer. The infiltration of fully expanded leaves of 4-week-old plants was carried out as described previously[59]. The density of the cultures was optimised to obtain sufficient HyPer2 expression for imaging 3–5 days after inoculation while avoiding the cytokinin-dependent side effects of Agrobacterium GV3101::pMP90 infiltration such as delayed leaf senescence, massively increased stromule formation and increased chloroplast-nucleus association[60].

**Live cell confocal imaging.** Three to five days after inoculation, 1 cm diameter leaf discs were cut and mounted in water. HL was applied to the discs mounted on the stage of a Nikon A1si confocal microscope equipped with a ×40 lens. The tungsten lamp, usually used for bright field illumination, was adapted for light treatments by adjusting the field diaphragm, to deliver a small, brightly illuminating circular beam[23]. Light intensity was measured with a light meter (SKP200, Skye, Powys, UK) and temperature with a homemade thermocouple. The intensity ratio image data were acquired using one-way sequential line scans of two excitation lines. For biosensors based on HyPer, the ratio of 488/405-nm laser power was kept constant at 1:3 and emission collected with one detector at 540/530 nm, with a photo-multiplier tube gain of 90–120 AU. No offset was used, and pinhole size was set between 1.2 and 2 times the Airy disk size of the used objective, depending on signal strength. Chloroplast stroma localisation of the biosensors was confirmed by chloroplast auto-fluorescence with excitation at 640 nm and emission at 670–720 nm. For the nuclear versions, the localisation was identified by correspondence with the characteristic size and shape of nuclei. To measure the effect of HL exposure on HyPer oxidation (indicated by increased 488/455 nm excitation ratio) it is necessary to turn off the HL source. A 2 min HL exposure is the minimum to allow photosynthesis to maintain the light-adapted trans-thylakoid proton gradient (Supplementary Fig. 3b) between exposures to the laser light for confocal ratio-metric imaging. Therefore the time resolution for measuring HyPer2 response to HL was 2 min.

**Chemical treatments.** H₂O₂, DTT and DCMU were obtained from Sigma-Aldrich. The leaf disks were kept for 20 min under illumination and floated on the following solutions. To drive HyPer2 to its oxidised and reduced forms we used 10 mM H₂O₂ and DTT. To confirm the lack of response to H₂O₂, cells expressing SypHer were incubated with 100 mM H₂O₂. HyPer2 F480/F405 nm fluorescence was measured after treatment of leaf discs with 10 μM DCMU for 20 min before HL exposure and imaging of epidermal cells.

**Image processing.** Images were analysed as described by Mishina et al.[61]. Briefly, images were smoothed using a Gaussian filter, and pixels containing the biosensor signal separated from the background using Otsu's method[62]. A ratio image was created by dividing the 488 nm image by the 405 nm image pixel by pixel and displayed using false colours.

**Statistical analysis.** For analysis of time course data for the responses of the F488/405 ratio of HyPer2 and SypHer2 in stroma, cytosol and nuclei the values were normalised to time zero. Due to the transient alkalinisation response in the first 2 min of exposure to HL and the effect this had on HyPer2 fluorescence, we disregarded the first time point after applying high light (2 min). This was because all the probes showed a transient increase in F488/405 ratio which interfered with curve fitting. To compare HL responses in each compartment we used GAMMs[63] to characterise treatment effects on the time series of Hyper2 and SypHer2 fluorescence in the nucleus, stroma and cytosol, while accounting for the hierarchical nature of our experimental data. For example, our experimental design yielded 5–70 replicate time series in each treatment for measurements made in the nucleus, stroma and cytosol. The hierarchical structure meant that measurements in each analysis were non-independent (i.e., time-series from the same replicate are likely to be auto-correlated). We accounted for this by treating replicates as a random effect on the intercept of the model, which models deviations among replicates from the fixed effects as normally distributed with a mean of zero. The most complex models included a treatment effect on the intercept (which characterises the median value of the response variable) and allowed the form of the time series, which was modelled using a cubic regression spline, to vary among treatments. Treatment effects on the form and intercept of the time series were modelled as fixed effects in the GAMMs. Model selection entailed fitting a range of models to the data, starting with the full model and then a series of reduced models with interaction terms and main effects removed to test hypotheses about the potential differences among treatments in the form of the time series.

For multi-model selection, we computed small sample-size corrected AIC scores (AICc) and then compared between models by calculating delta AICc values and AIC weights using the 'MuMIn' package. GAMMs were fitted to the data using the 'gamm4' package and all statistical analyses were conducted in R (v.3.23).

**Chlorophyll fluorescence imaging.** High-resolution images of the chlorophyll fluorescence (CF) parameter $F_v'/F_m'$ (PSII operating efficiency) in epidermal and mesophyll chloroplasts were obtained as described previously[64] using a Micro-FluorCam (Photon System Instruments, Czech Republic). Briefly, a single disc was placed onto the microscope slide, overlaid with a coverslip. The CF image data were collected over a range of PPFD (0, 100, 200, 300, 400, 500 μmol m⁻² s⁻¹). Using a 680 nm bandpass filter gives much greater weight to fluorescence from chloroplasts that are close to the imaged surface (including the chloroplasts within epidermal cells)[65]. $F_v'/F_m'$ was calculated according to Baker[66]. To assess the longer term effect of the HL treatment on photosynthesis, the dark-adapted $F_v/F_m$ (maximum PSII quantum efficiency)[65] was measured in HL-exposed *Nicotiana benthamiana* leaves after 30 min dark adaptation as previously described[11]. $F_v/F_m$ was calculated using the following equation $(F_m-F_o)/F_m$ using chlorophyll imaging system (Fluorimager, Technologica Ltd, Colchester, UK)[65]. Fo was measured during the weak measuring pulses of light (1 μmol m⁻² s⁻¹) to elicit a minimum value for chlorophyll fluorescence. $F_m$ was measured during an 800 ms exposure to a PPFD of approximately 4000 μmol m⁻² s⁻¹.

**APX activity.** Ascorbate peroxidase was measured according to Smirnoff and Colombé[67] by monitoring the rate of H₂O₂-dependent ascorbate oxidation followed at 280 nm ($\mathcal{E} = 6.15$ mol m⁻³ cm⁻¹) and scaled down for 96-well micro-plates. Agro-infiltrated leaf discs (100 mg) were homogenised in 0.5 ml of 50 mM potassium phosphate buffer pH 7, containing 1 mM EDTA, 0.1% Triton X 100 (v/v), 1% polyvinylpyrrolidone (w/v) and 1 mM sodium ascorbate. Extracts were centrifuged at 4 °C for 15 min at 16,000g, and the supernatants were analysed immediately for enzyme activity. The reaction mixture contained 50 mM potassium phosphate buffer pH 7, 1 mM EDTA, 0.4 mM ascorbate, 5 μl extract and 0.8 mM H₂O₂.

**RNA extraction and quantitative real-time RT-PCR.** RNA was extracted from 100 mg leaf disk using Triazol reagent (Sigma-Aldrich) according to the manu-facturer's instructions. RNA (5 ug) was treated with RNase-free DNase1 (Ambion). RNA (2 ug) was used to make random-primed cDNA except that the MuMLV reverse transcriptase was purchased from Fisher. Quantitative real-time PCR of the cDNA was performed using EvaGreen-fluorescence based procedure with reagents purchased from Applied Biological Materials (Canada). Relative and normalised fold expression values were calculated using the iQ5 Optical System Software v2.1 (Bio-Rad) normalised with respect to relative cDNA levels for *CYCLOPHILIN* (Niben101Scf01694Ctg023) and *ZIP9* (Niben101Scf03839Ctg048). The primers used in this study for quantitative RT-PCR are given in Supplementary Table 1.

**H₂O₂ extraction and quantification.** An Amplex Red Hydrogen Peroxide/Peroxidase Assay Kit (Invitrogen, Carlsbad, CA, USA) was used to measure H₂O₂ concentration in 4-week-old plants. Leaves were frozen in N₂ and then ground. Then 500 μl of phosphate buffer (20 mM K₂HPO₄, pH 6.5) was added to 50 mg of ground frozen tissue. After centrifugation, 50 μl of the supernatant was incubated with 0.2 U ml⁻¹ horseradish peroxidase and 100 μM Amplex Red reagent (10-acetyl-3,7-dihydrophenoxazine) at room temperature for 30 min in darkness. The fluorescence was quantified using a plate reader (FLUOStar Optima, BMG Labtech, Aylesbury, UK) with excitation at 560 nm and emission at 590 nm[68].

**Phylogenetic analysis.** To identify the putative orthologues of APX in *N. benthamiana*, we used the *A. thaliana* APX1 protein sequence in a BLAST search against the *N. benthamiana* genome (v0.4.4 predicted cDNAs) from the Sol Genomics Network (Cornell University; http://www.sgn.cornell.edu) and also against the peroxidase database PeroxiBASE (http://peroxibase.toulouse.inra.fr). MEGA[69] was used for the alignment of amino acid sequences and the inference of phylogenetic relationships. Aligned residues were used to generate a gene tree with the neighbour-joining method using 1000 samples for bootstrapping.

**Data availability.** The plasmids and sequence data based on HyPer2, sypHer2, pHRed, cAPX, sAPX that support the findings of this study have been deposited in Addgene (www.addgene.org) under the following codes: 84738 (pGem-sHyPer2: sAPX); 84737 (pGem-nHyPer2:sHyPer2); 84736 (pGem-HyPer2:cAPX); 84735 (pGem-nHyPer2:sAPX); 84734 (pUB-spHRed); 84733 (pUB-cSypHer2); 84732 (pUB-sSypHer2); 84731 (pUB-nSypHer2); 84730 (pUB-nHyPer2); 84729 (pUB-sHyPer2); 84728 (pUB-cHyPer2). The authors declare that all other data supporting the findings of this study are available from the corresponding authors on request.

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

## Acknowledgements

The research was funded by the UK Biotechnology and Biological Sciences Research Council (BBSRC, grant numbers BB/I020071/1 and BB/I020004/1). We are grateful to George R. Littlejohn for his input to initiating the research, advice on microscopy and discussion of the work.

## Author contributions

N.S. and P.M.M. conceived the investigation. M.E.R. and P.P.L. carried out the experimental work and, with G.Y.D., analysed the data. All authors contributed to manuscript preparation.

## Additional information

**Competing interests:** The authors declare no competing financial interests.

