## [Peer Review File · Nature Communications]

Reviewers' comments:

Reviewer #1 (Remarks to the Author):

This is a very interesting manuscript that uses state-of-the-art in vivo probes for hydrogen peroxide (H₂O₂) detection to study the transfer of H₂O₂ from the chloroplast to the nuclei during light stress. This is a highly novel study and the findings highlight a new principle in biology showing that only chloroplasts that are close to the nuclei participate in the transfer of H₂O₂ to the nuclei. The authors also use overexpression of stromal and cytosolic ascorbate peroxidases to show that the transfer of H₂O₂ is directly from the chloroplasts that are in direct contact with the nuclei to the nuclei. I find this paper to be highly novel and suitable for publication in Nature Communications.

I have only a few points of criticism:

1. The paper reads a bit too technical to me. I suggest to move more technical text to the method and/or supplemental material and to focus more on the science.
2. The method used for transformation and imaging includes agro infiltration and leaf discs. This is both a wounding response as well as a pathogen response. Have the authors tried generating stable tobacco plants? Just the wounding and the presence of the bacteria can skew the results because both are affecting ROS balance in the cell.
3. The authors should use TEM to investigate the type/mode of connection between the chloroplasts that are close to the nuclei and the nuclei. They indicate that they were not able to see stromules, but did they use the right tools, and maybe there are other structures that are novel?
4. A possibility that the authors did not consider is that a signal that is not H₂O₂ (but depends on H₂O₂ accumulation in the chloroplast) is transferred from the chloroplast to the nuclei and that this signal results in the enhanced production of H₂O₂ in the nuclei (via nuclei-specific ROS producing mechanisms).

Reviewer #2 (Remarks to the Author):

In the present study Exposito-Rodriguez and co-workers use the genetically-encoded fluorescent hydrogen peroxide sensor HyPer2 to demonstrate that high-light induced nuclear H₂O₂ accumulation is dependent on chloroplast H₂O₂ elevation and photosynthetic electron transport. The HyPer2 sensor is pH sensitive, and the authors therefore demonstrate that their signal is specific for redox by measuring pH with two independent pH sensors, pHRed and SypHer. Using stromal-targeted HyPer2 and pHRed, they show that stromal alkalization likely contributes to an initial, fast change in HyPer2 within the first five minutes of high-light treatment, but the overall increase in HyPer2 ratio at longer timescales is likely caused by oxidation and cannot be accounted for solely by pH changes. They also

perform parallel control experiment with the well-matched SypHer control sensor which is a redox-insensitive and pH-sensitive version of HyPer2. Overall, the use of HyPer2 and pH controls is detailed and careful, and it is convincing that the sensor signal is not solely caused by pH changes in most cases. However, there are some concerns which should be addressed:

Issues:

1) Figure 3F is too small and needs better explanation or labeling. Presumably the left column images are the overlays of HyPer fluorescence and chloroplast autofluorescence while the right column images are the HyPer ratio images?

2) One argument made by the authors argue that H₂O₂ is directly transferred from chloroplast to nucleus is that there is a fast correlated increase in nuclear and stromal HyPer2 signal. What is limiting the imaging interval to 2 minutes? It seems that a smaller imaging region around just the nucleus and associated chloroplasts or subset could be taken to image faster and test if indeed chloroplast oxidation precedes nuclear oxidation. For completeness, does nuclear-SypHer2 show any pH change?

3) The authors claim that direct transfer of H₂O₂ to the nucleus would mean very little H₂O₂ would diffuse into the cytosol from chloroplast. This implicates a unidirectional transfer mechanism, which would invoke some type of chloroplast-nuclear channel complex or diffusion restricted to the chloroplast-nuclear contact that is unknown because, as the authors discuss, very little is known about the physical connection. However, the ratio image in 3F seems to contradict this claim because it appears that there are indeed localized increases in cytosolic H₂O₂ in proximity to the three or four chloroplasts associated on the upper right portion of the nucleus. It would be helpful to address this more clearly and directly, also with an enlarged image as mentioned in point 1.

4) Related to the concern in 3, the authors discuss the steady-state lower HyPer2 ratio in chloroplasts associated with the nuclei compared to those not associated. Are the steady-state SypHer2 ratios consistent with this being due to H₂O₂ and not a steady-state pH difference? Is it possible to re-analyze data from Figure 2a to segregate nuclear and non-nuclear associated stromal signals to answer this question? Also, could there be a difference in antioxidant capacity for chloroplasts associated with the nuclei? Is it possible to re-analyze data from Figure 1 to segregate nuclear and non-nuclear associated stromal signals to answer this question, restated as: does H₂O₂ treatment cause a lower HyPer2 ratio in stroma associated with nuclei?

5) While the evidence is convincing that there is a chloroplast contribution to nuclear H₂O₂, there is no direct test if there is a peroxisomal contribution with targeted sensors and APX.

Reviewer #3 (Remarks to the Author):

The manuscript presents results on the tentative role of H₂O₂ in plastid signalling. A

biosensor system was used to localize H₂O₂ and the results obtained in combination with the use of different genotypes with altered scavenging and gene expression analyses allowed to conclude that H₂O₂ can leave the chloroplast. This confirms and massively extends a previous study (Mubarakshina et al. J Exp Bot. 2010) that has shown by EPR that H₂O₂ is produced during photosynthesis in the chloroplast and can leave this organelle. In particular, that a certain subpopulation of chloroplasts plays an important role in signalling is a completely new finding.

The strategy of the experiments presented in this manuscript is impressive. The manuscript is well written. Therefore, the study represents the most comprehensive set of experiments to date on this topic. A very intriguing conclusion is that generating plants that produce less H₂O₂ might be contra productive with respect to producing plant with enhanced acclimation capacity.

Details:

Summary. "Several potential operating signals originating from chloroplasts have been proposed, but none have been shown to transfer to nuclei to modulate gene expression." Is there a word missing? Shall this be "shown to transfer INFORMATION to nuclei"? Or a word less "shown to be transferred"?

Results. The use of the NbAPXa marker gene. Nowadays one would expect a RNA-Seq analysis to directly identify the perfect marker gene in *N. benthamiana* instead of simply relying on homologs from *Arabidopsis*.

Between them, the Reviewers have made some very useful comments which we have addressed below (in blue font). Where we have changed the manuscript this is indicated in bold font with line numbers.

Reviewers' comments:

Reviewer #1 (Remarks to the Author):

This is a very interesting manuscript that uses state-of-the-art in vivo probes for hydrogen peroxide (H₂O₂) detection to study the transfer of H₂O₂ from the chloroplast to the nuclei during light stress. This is a highly novel study and the findings highlight a new principle in biology showing that only chloroplasts that are close to the nuclei participate in the transfer of H₂O₂ to the nuclei. The authors also use overexpression of stromal and cytosolic ascorbate peroxidases to show that the transfer of H₂O₂ is directly from the chloroplasts that are in direct contact with the nuclei to the nuclei. I find this paper to be highly novel and suitable for publication in Nature Communications.

I have only a few points of criticism:

1. The paper reads a bit too technical to me. I suggest to move more technical text to the method and/or supplemental material and to focus more on the science.

Following editorial advice, we decided to leave the distribution of material unaltered.

2. The method used for transformation and imaging includes agro infiltration and leaf discs. This is both a wounding response as well as a pathogen response. Have the authors tried generating stable tobacco plants? Just the wounding and the presence of the bacteria can skew the results because both are affecting ROS balance in the cell.

We did not use stably transformed plants in this investigation because HyPer expression is strongly silenced soon after germination in Arabidopsis (Exposito-Rodriguez et al 2013). Also, the transient expression allowed us to rapidly introduce specific gene combinations as the research developed. The downside noted by the reviewer is that the agro-infiltration could induce wound and defence responses that potentially interact with the high light response. Agro-infiltration does have an effect and this is partly caused by synthesis of the plant cytokinin hormone zeatin by Agrobacterium (Erickson et al 2014, BMC Plant Biology 14, 127) leading to delayed chlorophyll loss (“green islands”), massive proliferation of stromules and increased clustering of chloroplasts around nuclei. However, these effects were caused by a high inoculum density (OD₆₀₀ ~0.8) while we optimised Agrobacterium density to OD 0.04 to 0.2 to obtain high HyPer expression without producing stromules and “green island” symptoms. The measurements of HyPer fluorescence were made 3-5 days after infiltration after which the initial bacterial responses would have damped down. Although an interaction with infiltration cannot be ruled out, the pattern of light-induced H₂O₂ formation we observe has been seen in non-infiltrated leaves (e.g. Murabakshina et al., 2010), so we are satisfied that our results reflect a normal response. **An explanation of the possible side effects and their mitigation has been added to Materials and Methods (lines 592 to 596).**

3. The authors should use TEM to investigate the type/mode of connection between the chloroplasts that are close to the nuclei and the nuclei.

We feel that it would not be fruitful to pursue the nature of chloroplast-nuclear connection in the context of this manuscript. Even if we were able to observe structures using TEM, this observation would be a long way from establishing functionality. In fact, the published TEMs (Selga et al) already show connections but alone this provides no functional information. To take this further would need an extensive investigation and the most likely approach would need extensive microscopy combined with identification of mutants affected in chloroplast-nucleus interactions to assess function.

They indicate that they were not able to see stromules, but did they use the right tools, and maybe there are other structures that are novel?

Like others, we have observed stromules in the context of response to pathogens but have not seen them in the HL response of the *N. benthamiana* epidermal cells. Therefore we think that our method is capable of detecting them if present since it is the same as other papers reporting stromules (i.e. chloroplasts labelled with fluorescent proteins and imaged by confocal microscopy). If there are “novel structures” the argument used in the first part of this question also applies.

4. A possibility that the authors did not consider is that a signal that is not H₂O₂ (but depends on H₂O₂ accumulation in the chloroplast) is transferred from the chloroplast to the nuclei and that this signal results in the enhanced production of H₂O₂ in the nuclei (via nuclei-specific ROS producing mechanisms).

The reviewer makes an important point and we did spend some time thinking about this issue. Nuclei could produce H₂O₂ from oxidase enzymes with flavin cofactors or by interaction of light with the flavin-containing blue light photoreceptor cryptochrome, which is located in nuclei. The involvement of cryptochrome is ruled out because it would not be affected by DCMU. There is not strong evidence for H₂O₂ generating proteins in nuclei based on current limited coverage of nuclear proteomes although, as expected if nuclear H₂O₂ is present, antioxidants such as glutathione and peroxiredoxins are present in nuclei. We therefore conclude that the currently most likely explanation for our results is that the chloroplast-sourced H₂O₂, which is released from chloroplasts closely associated with nuclei, is the same H₂O₂ that appears in the nuclei. **However, we do not have conclusive evidence and have added text to the discussion section to address this point (lines 393-401).**

Resolving this question would need very extensive work. Even if the nuclear H₂O₂ is produced in situ as a result of an H₂O₂-mediated signal from chloroplasts, then our essential conclusion that chloroplast-sourced H₂O₂ results in a nuclear H₂O₂ increase is not invalidated as long as we allow that it could be a two-step mechanism. Another point is the speed of response (see also Reviewer #2, point 2). Within a time resolution of 2 mins between measurements, there is no lag between chloroplast and nuclear H₂O₂ in response to high light (Fig. 2b and d). If production of nuclear H₂O₂ were a secondary response requiring, for example, activation of an enzyme then one might expect a detectable lag (whereas diffusion at this spatial scale would be in the millisecond to second range). We additionally expressed nuclear and chloroplast stroma HyPer simultaneously and were again unable to resolve a lag (Fig. 3b).

Reviewer #2 (Remarks to the Author):

In the present study Exposito-Rodriguez and co-workers use the genetically-encoded fluorescent hydrogen peroxide sensor HyPer2 to demonstrate that high-light induced nuclear H₂O₂ accumulation is dependent on chloroplast H₂O₂ elevation and photosynthetic electron transport. The HyPer2 sensor is pH sensitive, and the authors therefore demonstrate that their signal is specific for redox by measuring pH with two independent pH sensors, pHRed and SypHer. Using stromal-targeted HyPer2 and pHRed, they show that stromal alkalization likely contributes to an initial, fast change in HyPer2 within the first five minutes of high-light treatment, but the overall increase in HyPer2 ratio at longer timescales is likely caused by oxidation and cannot be accounted for solely by pH changes. They also perform parallel control experiment with the well-matched SypHer control sensor which is a redox-insensitive and pH-sensitive version of HyPer2. Overall, the use of HyPer2 and pH controls is detailed and careful, and it is convincing that the sensor signal is not solely caused by pH changes in most cases. However, there are some concerns which should be addressed:

Issues:

1) Figure 3F is too small and needs better explanation or labeling. Presumably the left column images are the overlays of HyPer fluorescence and chloroplast autofluorescence while the right column images are the HyPer ratio images?

The size of Fig. 3F is increased and the legend improved.

2) One argument made by the authors argue that H₂O₂ is directly transferred from chloroplast to nucleus is that there is a fast correlated increase in nuclear and stromal HyPer2 signal. **What is limiting the imaging interval to 2 minutes?** It seems that a smaller imaging region around just the nucleus and associated chloroplasts or subset could be taken to image faster and test if indeed chloroplast oxidation precedes nuclear oxidation.

The reviewer raises a very good point and we can add text to the Methods section to address it (lines 612-617). The reason for the two minute imaging interval is related to technical requirement needed to expose the leaf to high light while also obtaining the fluorescence data for the ratiometric imaging. Acquiring ratiometric images of HyPer while the sample is being subjected to HL is currently technically not feasible due to the different light sources for HL (white light) and ratiometric images (laser at 405/488 nm). We determined that the two minute interval (exposed HL) is the minimum to allow photosynthesis to maintain the light-adapted trans-thylakoid proton gradient (Suppl Fig 3B) between exposures to the laser light for confocal ratiometric imaging. Additionally, even if time better time resolution were technically possible, the relatively large variance in the early response to HL would make it very difficult to resolve a time lag.

For completeness, does nuclear-SypHer2 show any pH change?

There is no nuclear pH change (The data are in Fig 2D).

3) **The authors claim that direct transfer of H₂O₂ to the nucleus would mean very little H₂O₂ would diffuse into the cytosol from chloroplast.** This implicates a unidirectional transfer mechanism, which would invoke some type of chloroplast-

nuclear channel complex or diffusion restricted to the chloroplast-nuclear contact that is unknown because, as the authors discuss, very little is known about the physical connection. However, the ratio image in 3F seems to contradict this claim because it appears that there are indeed localized increases in cytosolic H₂O₂ in proximity to the three or four chloroplasts associated on the upper right portion of the nucleus. It would be helpful to address this more clearly and directly, also with an enlarged image as mention in point 1.

We think this query results from our poor description and we have modified the text to make the meaning clearer (lines 329-335). We did not intend to imply that H₂O₂ does not diffuse from nucleus-associated chloroplasts into the cytosol and, as pointed out by the reviewer, this does occur (Fig. 3F) and we now point this out more clearly. Therefore, with this re-wording, there is no need to invoke a unidirectional transport mechanism.

4) Related to the concern in 3, the authors discuss the steady-state lower HyPer2 ratio in chloroplasts associated with the nuclei compared to those not associated. Are the steady-state SypHer2 ratios consistent with this being due to H₂O₂ and not a steady-state pH difference? Is it possible to re-analyze data from Figure 2a to segregate nuclear and non-nuclear associated stromal signals to answer this question? Also, could there be a different in antioxidant capacity for chloroplasts associated with the nuclei? Is it possible to re-analyze data from Figure 1 to segregate nuclear and non-nuclear associated stromal signals to answer this question, restated as: does H₂O₂ treatment cause a lower HyPer2 ratio in stroma associated with nuclei?

This is an excellent point and it prompted us to analyse some additional images of the response of sSypHer to HL in chloroplasts associated with the nucleus compared to those distant. This showed a similar change in sSypHer fluorescence for both populations, so the oxidation state of Hyper does indeed differ between these chloroplast populations. Therefore, as suggested by the Reviewer, they could have different antioxidant capacities or indeed Mehler reaction capacity. These interesting possibilities could not be assessed without a major effort. **The new data are added to Fig 3C and described in lines 327-228.**

5) While the evidence is convincing that there is a chloroplast contribution to nuclear H₂O₂, there is no direct test if there is a peroxisomal contribution with targeted sensors and APX.

The reviewer makes an excellent point. We did point out that peroxisomes also produce H₂O₂ during photorespiration (e.g. line 265) but, as the reviewer says, we have no direct evidence for its contribution to nuclear H₂O₂. We argue that our data show clearly that a large proportion of nuclear H₂O₂ is chloroplast-sourced. As described in the manuscript, expression of the H₂O₂ scavenging enzyme ascorbate peroxidase (APX) in chloroplast stroma or cytosol shows that only the former has an effect on nuclear H₂O₂ or on expression of the light responsive target gene (NbAPXa). There is no evidence that peroxisomes are attached to nuclei but a sub-population is attached to chloroplasts (Gao et al 2016, Plant Physiology 170, 263), so peroxisome-sourced H₂O₂ would need to transit the cytosol. Given that cytosolic APX had no effect on nuclear H₂O₂ and gene expression, we conclude that peroxisomes are a minor source. Less conclusively, but in support, although oxidised HyPer could be seen around chloroplasts, we never saw spots of oxidation of cytosolic HyPer that could have been associated with the (much smaller)

peroxisomes. **The text has been revised to make these points clearer (lines 281, 423 and 459-463).**

Reviewer #3 (Remarks to the Author):

The manuscript presents results on the tentative role of H₂O₂ in plastid signalling. A biosensor system was used to localize H₂O₂ and the results obtained in combination with the use of different genotypes with altered scavenging and gene expression analyses allowed to conclude that H₂O₂ can leave the chloroplast. This confirms and massively extends a previous study (Mubarakshina et al. J Exp Bot. 2010) that has shown by EPR that H₂O₂ is produced during photosynthesis in the chloroplast and can leave this organelle. In particular, that a certain subpopulation of chloroplasts plays an important role in signalling is a completely new finding. The strategy of the experiments presented in this manuscript is impressive. The manuscript is well written. Therefore, the study represents the most comprehensive set of experiments to date on this topic. A very intriguing conclusion is that generating plants that produce less H₂O₂ might be contra productive with respect to producing plant with enhanced acclimation capacity.

Details:

Summary. “Several potential operating signals originating from chloroplasts have been proposed, but none have been shown to transfer to nuclei to modulate gene expression.” Is there a word missing? Shall this be “shown to transfer INFORMATION to nuclei”? Or a word less “shown to be transferred”? **Summary text has been revised. We are at the word limit and this might need more attention if not satisfactory.**

Results. The use of the NbAPXa marker gene. Nowadays one would expect a RNA-Seq analysis to directly identify the perfect marker gene in *N. benthamiana* instead of simply relying on homologs from *Arabidopsis*. **We could have carried out an mRNA-seq experiment to identify marker genes but in this instance it actually proved effective to identify one from the detailed knowledge of HL/H₂O₂ responsive genes in *Arabidopsis*. For further work it would make sense to take a transcriptome approach but it is not relevant to the current manuscript.**

REVIEWERS' COMMENTS:

Reviewer #1 (Remarks to the Author):

The authors have answered all of the points I felt needed to be answered to merit publication.

Reviewer #2 (Remarks to the Author):

- 1) The figure legend for 3F is clarified.
- 2 & 3) The analysis and discussion of direct chloroplast to nuclear transfer of H₂O₂ is clarified in conjunction with the response to Reviewer 1's Point 3&4, regarding the need for future work to define the molecular nature of the chloroplast-nuclear physical coupling and alternative signaling paths.
- 4) The analysis added in 3C addresses the concerns. Investigation of the potential difference in antioxidant capacity will be interesting in future work.
- 5) The discussion clarifies this concern.

The authors provided a detailed and thorough response to all the concerns raised initially. The manuscript, in both iterations, was very interesting and enjoyable to read.

Reviewer #3 (Remarks to the Author):

The authors have addressed my few points of criticism. As far as I can see, they have also done a good job in responding to the more extensive comments of the other two reviewers. The revised version is in a very good shape and of high interest for the field of intracellular signaling in plants and beyond. Congratulations to this nice work.